# Growth Charts for Shwachman–Diamond Syndrome at Ages 0 to 18 Years

**DOI:** 10.3390/cancers16071420

**Published:** 2024-04-05

**Authors:** Anna Pegoraro, Valentino Bezzerri, Gloria Tridello, Cecilia Brignole, Francesca Lucca, Emily Pintani, Cesare Danesino, Simone Cesaro, Francesca Fioredda, Marco Cipolli

**Affiliations:** 1Cystic Fibrosis Center, Azienda Ospedaliera Universitaria Integrata, 37126 Verona, Italy; anna.pegoraro@aovr.veneto.it (A.P.); v.bezzerri@unilink.it (V.B.); gloria.tridello@aovr.veneto.it (G.T.); cecilia.brignole@aovr.veneto.it (C.B.); francesca.lucca@aovr.veneto.it (F.L.);; 2Department of Life Sciences, Health, and Health Professions, Link Campus University, 00165 Rome, Italy; 3Department of Molecular Medicine, University of Pavia, 27100 Pavia, Italy; cidi@unipv.it; 4Pediatric Hematology Oncology, Azienda Ospedaliera Universitaria Integrata, 37126 Verona, Italy; simone.cesaro@aovr.veneto.it; 5Unit of Hematology, IRCCS G. Gaslini Hospital, 16147 Genoa, Italy; francescafioredda@gaslini.org

**Keywords:** Shwachman–Diamond syndrome, bone marrow failure, pancreatic insufficiency, ribosomopathy

## Abstract

**Simple Summary:**

In this study, we drew up the growth charts of Italian patients with Shwachman–Diamond syndrome (SDS) at ages 0 to 18 years. We found that the 50th and 3rd percentiles of weight and height of the pediatric general population correspond to the 97th and 50th percentiles of patients with SDS, respectively. The median age at menarche in females with SDS was comparable with that of the general population. The percentage increment in weight of subjects aged 14–18 years was higher in patients with SDS than in the general population. This study provides insight into the potential usefulness of SDS-specific growth chart data as a resource for clinicians working with patients with SDS.

**Abstract:**

Shwachman–Diamond syndrome (SDS) is one of the most common inherited bone marrow failure syndromes. SDS is characterized by hypocellular bone marrow, with a severe impairment of the myeloid lineage, resulting in neutropenia, thrombocytopenia, and, more rarely, anemia. Almost 15% of patients with SDS develop myelodysplastic syndrome or acute myeloid leukemia as early as childhood or young adulthood. Exocrine pancreatic insufficiency is another common feature of SDS. Almost all patients with SDS show failure to thrive, which is associated with skeletal abnormalities due to defective ossification. Considering these observations, it remains unfeasible to use the common growth charts already available for the general population. To address this issue, we report how we drew up growth charts of patients with SDS aged 0 to 18 years. We analyzed height, weight, and body max index (BMI) in 121 Italian patients with SDS. Results indicated that the 50th and 3rd percentiles of weight and height of the pediatric general population correspond to the 97th and 50th percentiles of patients with SDS aged 0–18 years, respectively. In addition, the percentage increment in weight of subjects aged 14–18 years was higher in patients with SDS than in the general population. SDS-specific growth charts, such as those described here, afford a new tool, which is potentially useful for both clinical and research purposes in SDS.

## 1. Introduction

Shwachman–Diamond syndrome (SDS) is the third most common inherited bone marrow failure syndrome (IBMFS) [1,2]. Almost 95% of patients with SDS carry biallelic mutations in the Shwachman–Bodian–Diamond Syndrome (*SBDS*) gene [3]. SBDS protein is involved in ribosome assembly, collaborating with Elongation Factor Like GTPase 1 (EFL1) in the release of Eukaryotic Translation Initiation Factor 6 (eIF6) factor from the eukaryotic large S60 ribosomal subunit [4,5,6]. SDS has, indeed, been classified as a ribosomopathy [7,8]. Skeletal malformations, due to an abnormal ossification process, cause failure to thrive even at an early stage of life in patients with SDS [9]. The primary skeletal defect associated with the lack of functional SBDS protein is abnormal development of bone metaphysis with delayed secondary ossification centers, particularly in the femoral and humeral heads, knees, ankles, and rib cage [10]. Biopsies from the iliac crests showed reduced trabecular bone and osteoid in the absence of mineralization deficiency [11].

SDS is characterized by bone marrow failure and is expressed mainly by neutropenia rather than thrombocytopenia. This condition also involves a high risk of myelodysplastic syndrome (MDS) and acute myeloid leukemia (AML) [12,13]. Exocrine pancreatic insufficiency is another feature of this disease. Pancreatic dysfunction is due to pancreatic acinar cell atrophy followed by adipocyte replacement, in the absence of appreciable inflammation [14]. Of note, a recent study of *Sbds* mutant zebrafish phenocopying the SDS condition has shown that mutant larvae exhibit hepatocyte and pancreatic lobular exocrine atrophy, with reduced size of zymogen granules [15]. Consistent with these findings, patients with SDS need oral pancreatic enzyme supplementation and monitoring of their nutritional status. However, in most patients with SDS, pancreatic function improves over time, showing increased levels of lipase, amylase, trypsin, and chymotrypsin [16]. Given these premises, malnutrition and malabsorption of nutrients should be considered as the aggravating factors of failure to thrive, but not causative [17].

According to the WHO standard references [18,19], we already reported that the 50th and 3rd percentiles of weight and height of the pediatric general population correspond to the 97th and 50th percentiles, respectively, of patients with SDS aged 0–8 years, whereas no difference was observed in body mass index (BMI) [20].

In this study, we extend the previously reported growth charts (GCs) to include patients with SDS up to the age of 18 years, as a new reference tool for evaluating the nutritional aspects of SDS. Additionally, we included the effect of menarche on the growth of female patients.

## 2. Materials and Methods

### 2.1. Patient Recruitment

We conducted a retrospective observational study on patients included in the Italian SDS Registry, born between 1969 and 2019. One hundred and twenty-one patients included in the registry, with 707 total measurements, were considered eligible for this study. The data items collected were sex, birth date, height, weight, and clinical information at follow-up assessment, from birth to the age of 18 years.

### 2.2. Statistical Analysis

The primary endpoint was the evaluation of the percentiles for height, weight, and BMI for males and females with SDS until adulthood. The LMS method, developed by Cole and Green as a standard means of estimating percentiles and generating growth charts, was used here: this method allows the generation of reference charts that show the distribution of a measurement as it changes, in relation to covariates such as age and time. The LMS parameters describe the degree of skewness (L for lambda), the median (M for mu), and the coefficient of variation (S for sigma) for each measurement [21,22]. This method can also be used for the calculation of percentiles, as SD scores can be converted to percentiles in a normal distribution. Worm plots were used to verify the difference of the expected and the empirical standardized residuals. The Q-Q plot was used to verify the normal distribution of residuals. The LMS method, which includes smoothing with a spline function after normalizing age-specific data through a power transformation, was used to derive the LMS parameters from the weight and height data.

A 3-month window was used for each of the three anthropomorphic measurements; in case of multiple assessments, the mean of the values available inside the window was considered valid. Growth curves for weight, height, and BMI from birth to 18 years were constructed, for males or females, each with 3rd, 25th, 50th, 75th, and 97th percentiles for age.

The secondary endpoints were (i) comparison of the estimated growth charts of patients with SDS with the general population and (ii) identification of age at menarche. These parameters were analyzed using descriptive statistics: median, minimum, and maximum values were used to summarize continuous variables, whereas absolute and percentage frequencies were used for categorical variables. The Cacciari percentiles [23], available for people aged 2–18 years, were used to generate the GC of the general population in this age range; for ages 0–2 years, the WHO reference [24] was used. The percentage change was obtained by calculating the difference at the 50th percentiles of the specific parameter between the highest and lowest ages, divided by the value at the lowest age, and multiplied by 100.

The analysis was performed by means of the statistical software SAS (ver. 9.4) and the “gamlss” package of the software R, Version 4.3.1. (R Project for Statistical Computing).

## 3. Results

### 3.1. Study Population

This study included 121 patients (51 females, 42.1%) (Table 1), totaling 432 and 275 observations for males and females, respectively (Appendix A). All patients with pancreatic insufficiency were undergoing pancreatic enzyme replacement therapy (PERT).

The median number of observations per patient was 5 (min–max 1–21). Median age at diagnosis was 1.3 years (min–max 0–35.6), while gestational age was 39 weeks, (min–max 29–42 weeks); median weight at birth was 2700 g (900–4200 g).

The most frequent combination of mutations in *SBDS* was the c.258+2T>C splicing variant with the c.183-184TA>CT nonsense mutation, which was observed in 67 (55.4%) patients. Most patients had pancreatic insufficiency and were receiving pancreatic enzyme treatment at the most recent follow-up. Eight patients (6%) were undergoing growth hormone (GH) therapy, starting at a median age of 11.0 years (2.9–15.4), with only one patient still receiving GH at the time of analysis. At a median age of 6.4 years (min–max 2.2–36.8), 19 patients (16.7%) underwent HSCT due to MDS (6), aplasia (6), leukemia (5), or for unknown reasons (2). The median age at menarche in females with SDS was 12 years (11–15), which is comparable with that of the Italian general population (12.4 years (95%CI, 12.4; 12.6), as previously described [25]. It was not possible to retrieve the age at pubarche.

### 3.2. SDS Percentiles and Growth Charts

The 3rd, 25th, 50th, 75th, and 97th percentiles for height, weight, and BMI are reported in Appendix A. The corresponding growth charts were generated for males and females (Figure 1).

To compare GCs for patients with SDS and the Italian general population, we show the charts for both populations in Figure 2. The comparison shows that the 97th and 50th percentiles for height and weight of patients with SDS correspond to the 50th and 3rd percentiles of the general population, respectively. This trend is maintained throughout the observation period, up to 18 years of age. The GCs for the BMI were quite similar between the two populations, with the only exception being that the 97th percentile remains clearly lower for patients with SDS.

### 3.3. Percentage Changes in Height, Weight, and BMI

To further investigate the difference in height, weight, and BMI increases between patients with SDS and the general population, we calculated the percentage increase in each of the 4-year age brackets from 2 to 18 years. Since the percentiles for the Italian general population are not available in people aged 0–2 years, we performed the comparison starting from the age of 2 years.

Table 2 shows the percentage change of 50th percentiles for the three parameters, for males and females separately. The percentage change was calculated as follows, taking the 50th percentile for height in males with SDS as an example: difference in the 50th percentile between 6 years and 2 years of age, divided by the 50th percentile at 2 years, and multiplied by 100 = (106.32 − 79.74)/79.74 × 100. This showed a 33.3% increase from 2 to 6 years.

The percentage of height increment between the patients with SDS and the general population is comparable in all age categories, including the difference between 10 and 14 years for females, when the menarche occurs (Table 2). In contrast, the percentage in weight increment of subjects aged 14–18 years was higher in patients with SDS than in the general population, showing a ~2-fold increase and a ~8-fold increase for males and females, respectively. This resulted in a 3.5-fold increase and 7.5-fold increase in BMI for males and females with SDS, respectively, within the same age bracket.

## 4. Discussion

We previously reported the growth charts of Italian patients with SDS aged 0–8 years, showing that the 50th and 3rd percentiles of weight and height of the Italian general population correspond to the 97th and 50th percentiles of the SDS population, respectively [20]. This study strengthens our previous observation, extending the analysis to a total of 121 subjects aged up to 18 years and included in the Italian SDS Registry. All patients enrolled in this study were diagnosed with SDS through a genetic analysis and showed biallelic mutations in *SBDS*. Patients carrying *DNAJC21*, *SRP54*, and *EFL1* variants, described as subjects with an SDS-like condition [14], were not included in this study. Considering the rarity of SDS [26,27], this study’s population is one of the largest cohort studies ever reported for this disease.

Consistent with the previous study on patients with SDS in childhood, the 97th and 50th percentiles for height and weight of patients with SDS aged 0–18 years still correspond to the 50th and 3rd percentiles of the general population, respectively. In addition, we found that the median age at menarche of patients with SDS (12 years) was comparable with that of the Italian general population (12.4 years) [14]. In this regard, we observed that pubertal development did not modify the growth trend in females with SDS. Unfortunately, we were not able to evaluate age at pubarche. Nevertheless, considering the already available data for adult patients with SDS and looking at our results, it seems reasonable to posit tentatively that the growth trend in males with SDS is similarly unaffected by pubertal development.

This is the first study reporting growth charts for patients with SDS until adulthood. It has a number of limitations that should be considered. First, the patients enrolled in this study generally received medical care and thus do not represent the natural development of the disease. In addition, GCs are normally generated using data collected longitudinally over an extended period in large groups [28]. However, the nature of our cohort did not allow us to use this method. Thirdly, our data were collected on the Italian population, which may at least partially overlap with the Caucasian population: we cannot exclude that some differences may be observed when studying other ethnic groups.

In SDS, pancreatic insufficiency is generally associated with gastrointestinal symptoms and malnutrition [17]. Malabsorption of nutrients, including vitamins D and K, contributes to osteoporosis, affecting almost all patients with SDS [9]. However, the failure to thrive displayed by patients with SDS should be considered as a consequence of the basic genetic defect, rather than malnutrition. Despite their short stature, the BMI of patients with SDS falls within the normal range in childhood and adolescence [20]. Here, we observe that patients with SDS, and particularly females in the age bracket from 14 to 18 years, show a higher percentage increase in weight and BMI than the general population. This corroborates the hypothesis that failure to thrive is not linked to malnutrition, but to the mutant *SBDS*-dependent inhibition of osteogenesis. The weight gain observed in patients with SDS aged 14–18 years may reflect lower social pressure to take care of their physical shape as a result of their chronic disease. Another possible explanation for the greater weight gain in patients with SDS than in the general population is that most of them are less physically active due to skeletal abnormalities and joint pain.

## 5. Conclusions

In conclusion, the extended growth charts reported in this study can potentially provide an excellent tool for clinicians dealing with SDS, facilitating more accurate follow-ups of young patients with SDS. Additionally, these growth charts may assist investigators developing innovative therapies aimed at restoring the basic SDS defect [5,29,30], which are currently eliciting growing interest in the SDS scientific community.

## Figures and Tables

**Figure 1 cancers-16-01420-f001:**
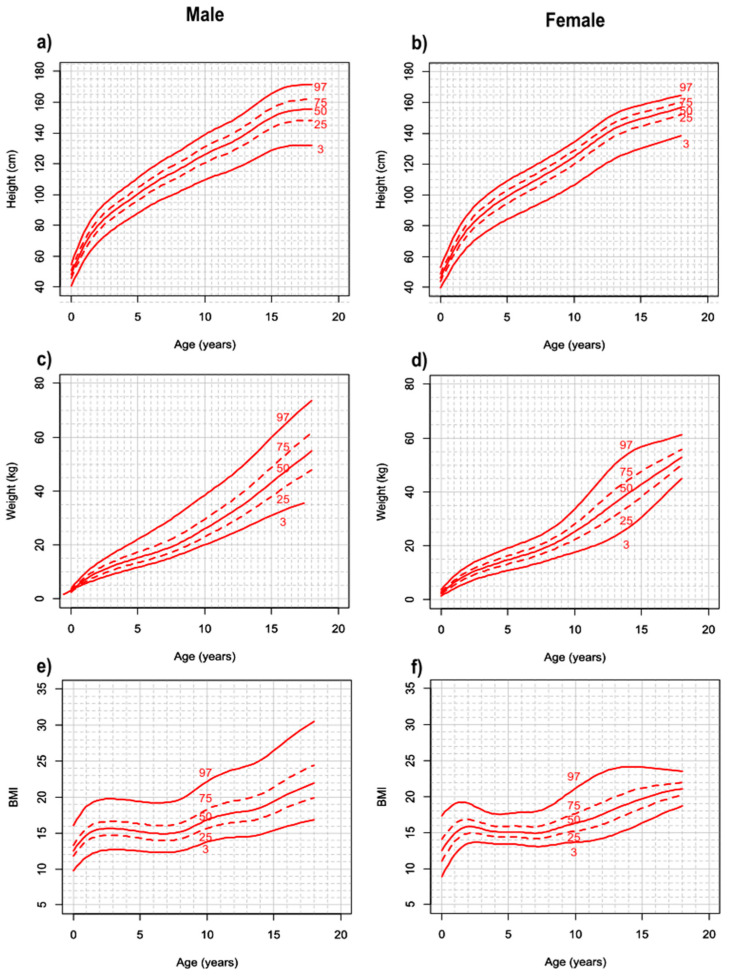
Growth charts for height, weight, and body mass index (BMI) of patients with SDS at ages 0 to 18 years. Growth charts for height (**a**,**b**), weight (**c**,**d**), and BMI (**e**,**f**) in males (**left**) and females (**right**) with SDS are shown.

**Figure 2 cancers-16-01420-f002:**
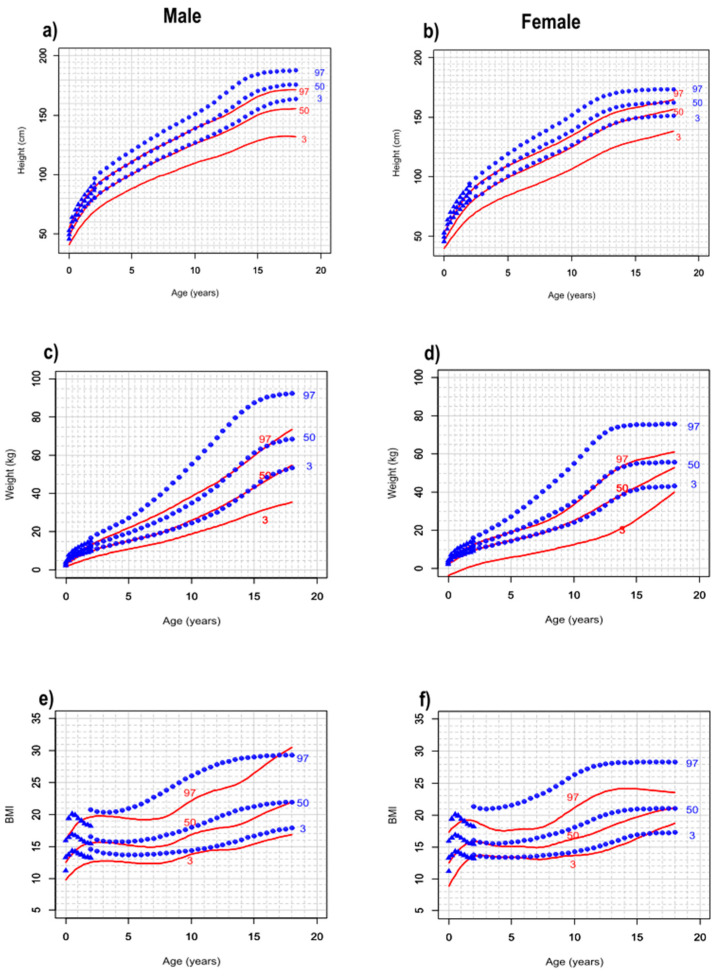
Comparison of growth charts between patients with SDS and the Italian general population at ages 0 to 18 years. The plots show the comparisons of growth charts for height (**a**,**b**), weight (**c**,**d**), and BMI (**e**,**f**) between patients with SDS (continuous red lines) and the Italian general population (blue dotted lines).

**Table 1 cancers-16-01420-t001:** Study population characteristics.

Variables	N
Total	121
Sex	
Males (%)	70 (57.9)
Females (%)	51 (42.1)
Age at diagnosis (months)	
Median (min–max)	1.3 (0–35.6)
Gestational age (weeks)	
Median (min–max)	39 (29–42)
Weight at birth	
Median (min–max)	2.7 (0.9–4.2)
Pancreatic insufficiency at follow-up	
Yes (%)	80 (66.1)
No (%)	31 (25.6)
Unknown (%)	10 (8.3)
Pancreatic insufficiency (at least once)	
Yes (%)	113 (93.4)
No (%)	7 (5.8)
Unknown (%)	1 (0.8)
Pancreatic enzyme	
No, never (%)	10 (8.3)
Yes, ongoing (%)	80 (66.1)
Yes, in the past, but stopped (%)	22 (18.2)
Unknown (%)	9 (7.4)
GH	
Yes (%)	8 (6.6)
No (%)	113 (93.4)
HSCT	
Yes (%)	19 (15.7)
No (%)	102 (84.3)
Mutations	N (%)
c.258+2T>C and c.183-184TA>CT	68 (56.2)
c.258+2T>C and c.258+2T>C	12 (9.9)
c.258+2T>C and c.183-184TA>CT+c.258+2T>C	17 (14)
c.258+2T>C and c.258+533_459+403del	5 (4.1)
c.258+2T>C and c.187G>T	2 (1.7)
c.258+2T>C and c.101A>T	1 (0.8)
c.258+2T>C and c.107delT	1 (0.8)
c.258+2T>C and c.128+6T>C	1 (0.8)
c.258+2T>C and c.212C>T	1 (0.8)
c.258+2T>C and c.289-292del	1 (0.8)
c.258+2T>C and c.300delAC	1 (0.8)
c.258+2T>C and c.307-308delCA	1 (0.8)
c.258+2T>C and c.352A>G	1 (0.8)
c.258+2T>C and c.356G>A	1 (0.8)
c.258+2T>C and c.624+1G>C	1 (0.8)
c.258+2T>C and c.92-93GC>AG	1 (0.8)
c.258+2T>C and c.184A>T	1 (0.8)
c.258+2T>C and c.18dup	1 (0.8)
c.258+2T>C and c.IVS1-71del83bp	1 (0.8)
c.258+2T>C and c.652C>T	1 (0.8)
c.258+2T>C and c.95A>G	1 (0.8)
c.523C>T and c.523C>T	1 (0.8)

GH, growth hormone; HSCT, hematopoietic stem cell transplantation.

**Table 2 cancers-16-01420-t002:** Percentage of increase in height, weight, and BMI between patients with SDS and general population.

Age	SDS M	SDS F	Gen. Pop. M *	Gen. Pop. F *
Height				
2–6	33.3	34.8	31.5	33.1
6–10	18.5	19.9	19.0	19.7
10–14	14.8	17.6	18.4	14.9
14–18	7.5	6.8	6.9	1.8
Weight				
2–6	71.1	73.7	71.9	78.0
6–10	55.0	55.0	59.5	60.7
10–14	50.7	56.6	59.0	52.6
14–18	40.2	34.0	23.1	4.1
BMI				
2–6	−3.6	−4.9	−3.0	0.0
6–10	12.4	8.5	12.5	12.4
10–14	10.1	16.7	16.1	14.9
14–18	18.5	10.7	5.3	1.4

* Italian general population, as described by Cacciari et al. [23]. (M, males; F, females).

## Data Availability

The raw data supporting the conclusions of this article are partially included in Appendix A. The remaining data (patient information) will be made available by the authors upon request.

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
