# Peer review of "Growth Charts for Shwachman–Diamond Syndrome at Ages 0 to 18 Years"

_cancers, 2024, doi:10.3390/cancers16071420_

Round 1
Reviewer 1 Report
Comments and Suggestions for Authors
Dear Authors,
Thank you very much for submitting such an interesting communication to the Cacners. Below I present point-by-point comments and suggestions regarding your paper:
- A simple summary should contain only the most crucial information. In this form, information included in the simple summary should rather be in the abstract section. In this form some of the important information is in the abstract while some in the simple summary. Add methodology and results in the abstract while general information in the simple summary Please correct it.
- Please add the abbreviations section
Comments on the Quality of English LanguageIn the text there are several minor/major grammatical mistakes that should be corrected
Author Response
Thank you very much for submitting such an interesting communication to the Cancers. Below I present point-by-point comments and suggestions regarding your paper:
R1.1. We are grateful for considering our manuscript interesting.
- A simple summary should contain only the most crucial information. In this form, information included in the simple summary should rather be in the abstract section. In this form some of the important information is in the abstract while some in the simple summary. Add methodology and results in the abstract while general information in the simple summary Please correct it.
R1.2. We edited the simple summary according to these observations. We added the number of patients enrolled in this study in the abstract.
- Please add the abbreviations section
R1.3. Please note that we followed the Cancers instructions for authors, where it is specified that “Abbreviations/Initialisms should be defined the first time they appear in each of three sections: the abstract; the main text; the first figure or table. When defined for the first time, the acronym/abbreviation/initialism should be added in parentheses after the written-out form”. However, we have checked the abbreviations throughout the text in the revised version and edited the text accordingly. We are ready to provide any further details that may be required.
In the text there are several minor/major grammatical mistakes that should be corrected
R1.4. The text has now been internally reviewed by a native English speaker colleague.
Reviewer 2 Report
Comments and Suggestions for Authors
This communication presents growth charts for the Italian population for the age bracket 0 to 18 of SDS patients. The is short and present major concerns.
Despite the data being potentially interesting and useful, it has low relevance to cancer, which is the subject of the present journal.
The study could do more in regard to statistics, and stress more the statistically significant differences compared to the general population.
Considering the low relevance to the journal, and the little depth given to the statistical analysis, I recommend against proceeding with the publication on Cancers.
Author Response
This communication presents growth charts for the Italian population for the age bracket 0 to 18 of SDS patients. The is short and present major concerns.
Despite the data being potentially interesting and useful, it has low relevance to cancer, which is the subject of the present journal.
R2.1. We thank this Reviewer for considering our data potentially interesting and useful. We understand the point of view expressed by this Reviewer. However, we submitted the manuscript for the special issue of Cancers entitled “Recent Advances in the Understanding of Myelodysplastic Syndrome and Acute Myeloid Leukemia”. Our growth charts may represent a useful tool for the management of SDS, which is one of the most common inherited bone marrow failure syndromes, associated with MDS and AML. In addition, we previously submitted an inquiry to the editors before submitting the full manuscript, receiving editorial approval for submission.
The study could do more in regard to statistics, and stress more the statistically significant differences compared to the general population. Considering the low relevance to the journal, and the little depth given to the statistical analysis, I recommend against proceeding with the publication on Cancers.
R2.2. We improved the statistical analysis session in Methods, according to the Reviewer’s recommendations. The comparison of growth charts between SDS patients and the general population was included, to highlight the actual differences and to provide a specific tool to evaluate the change of height, weight, and BMI over time in the SDS population.
Reviewer 3 Report
Comments and Suggestions for Authors
The authors reported one of the largest cohort of patients with SDS. Given the rarity of this condition, this study represents an important guide for clinicians.
The study is an expansion of a previous one published by the authors and aims to describe the growth chart of Italian patients with SDS aged up to 18 years. It is valid and written clearly with a direct message.
The authors indicate also the limitations of their study.
Author Response
The authors reported one of the largest cohort of patients with SDS. Given the rarity of this condition, this study represents an important guide for clinicians.
R3.1. We thank this Reviewer for considering our manuscript as an important guide for clinicians. This is our point of view as well. We agree on the importance of such an enlarged cohort of patients, which was made possible thanks to the Italian Registry of SDS patients.
The study is an expansion of a previous one published by the authors and aims to describe the growth chart of Italian patients with SDS aged up to 18 years. It is valid and written clearly with a direct message.
The authors indicate also the limitations of their study.
R3.2. Again, we thank this Reviewer for appreciating our study.
Round 2
Reviewer 2 Report
Comments and Suggestions for Authors
I appreciate the authors' effort in addressing the points I have raised. As my doubts are clarified and the concerns have been addressed, I see no further problems with the manuscript's current status.